# Local Disentanglement in Variational Auto-Encoders Using Jacobian $L_1$ Regularization

**Travers Rhodes**
Department of Computer Science
Cornell Tech, Cornell University
New York, NY 10044
tsr42@cornell.edu

**Daniel D. Lee**
Department of Electrical and Computer Engineering
Cornell Tech, Cornell University
New York, NY 10044
ddl46@cornell.edu

## Abstract

There have been many recent advances in representation learning; however, unsupervised representation learning can still struggle with model identification issues related to rotations of the latent space. Variational Auto-Encoders (VAEs) and their extensions such as $\beta$-VAEs have been shown to improve local alignment of latent variables with PCA directions, which can help to improve model disentanglement under some conditions. Borrowing inspiration from Independent Component Analysis (ICA) and sparse coding, we propose applying an $L_1$ loss to the VAE's generative Jacobian during training to encourage local latent variable alignment with independent factors of variation in images of multiple objects or images with multiple parts. We demonstrate our results on a variety of datasets, giving qualitative and quantitative results using information theoretic and modularity measures that show our added $L_1$ cost encourages local axis alignment of the latent representation with individual factors of variation.

## 1 Introduction

Unsupervised representation learning takes a collection of image data from the world and figures out how to organize and find patterns in the data without additional information about how the images were generated. The ideal representation learning algorithm would compress high-dimensional image data into a lower-dimensional latent representation that contains relevant information about the ground-truth factors of variation that generated the image.

Inferring a good latent representation from a dataset is a difficult problem and is generally underspecified in algorithms. This underspecification is called the "model identification" problem, and one example is the fact that representation learning algorithms often struggle to precise the correct orientation of a latent space. That is, optimization criteria used to learn a representation function might be equally well satisfied by an equivalent representation that is just a rotation of the latent space by an arbitrary amount.

As commonly implemented (using axis-aligned Gaussian posterior distributions), Variational Auto-Encoders (VAEs) [1] and their extension, the $\beta$-VAE [2], solve the rotational part of the model identification issue by tending to ensure that the generation function's Jacobian matrix has orthogonal columns (i.e., that the generative Jacobian matrix's right singular values are aligned with the axes of the latent space) [3, 4]. Intuitively, this is because the VAE's stochastic reconstruction cost prefers to budget higher precision (lower embedding noise) in the directions along which the generative Jacobian changes most rapidly. Kumar and Poole [4] draw the parallel between this preferred orientation and linear Principal Component Analysis (PCA). However, we note that learning algorithms that resolve rotational identification issues through methods related to PCA will still suffer from an identifiability issue related to rotations that mix the directions for which the generative Jacobian matrix has equal

singular values. Along these directions, the posterior Gaussian would have approximately equal variance and be rotationally symmetric.

We propose adding an $L_1$ cost to the generation function's Jacobian matrix as a way to resolve that rotational identifiability issue. Since $L_1$ cost is not rotation invariant, $L_1$ regularization creates a preferred latent-space orientation among directions whose singular values are equal. This use of the $L_1$ norm to choose an orientation is inspired by similar use in linear models. For example, when using a Laplacian prior in Independent Component Analysis (ICA) [5] and applying it to already-whitened data, ICA rotates the data to minimize the $L_1$ norm. As shown in Olshausen [6], that ICA formulation is equivalent to sparse coding using an $L_1$ cost [7], with the same rotational effect. In Sparse PCA, the $L_1$ norm encourages sparsity in the loadings/mixing matrix (rather than the principal components/sources as in ICA) [8, 9], similarly encouraging preferred orientations. These techniques for preferring certain orientations using the $L_1$ norm are all linear model techniques, and we apply them to non-linear VAEs by regularizing the $L_1$ norm of the generator Jacobian matrix, thereby encouraging a preferred orientation for our non-linear models.

The motivation above suggests that an $L_1$ norm on the generator Jacobian can address the rotational identifiability issue, and we think the $L_1$ regularization will encourage useful orientations of the latent space for image data. This belief comes from results on the sparse linear coding of images. In particular, Olshausen and Field [7] showed how sparse linear coding on natural images generates local receptive fields similar to those discovered in the mammalian visual processing system. These types of localized receptive fields are shown in Figure 1. Each image in that figure corresponds to a direction in the latent space, and shows how perturbing the latent value in that direction changes the generated image—white pixels means the image gets brighter there, black pixels means the image gets darker. For the ICA basis, we see that perturbations in different latent directions are associated with localized changes to the image, whereas for the PCA basis, the latent directions tend to affect the entire image. Adding an $L_1$ penalty to our model should encourage sparsity within the generative Jacobian columns of our model, meaning that perturbations of latent values along individual latent directions should modify as few pixels as possible, leading to latent directions that affect the output image in localized regions. $L_1$ generative Jacobian regularization should therefore disentangle representations of different objects in an image. We call the proposed model trained with this additional regularization a Jacobian $L_1$ Regularized Variational Auto-Encoder (JL1-VAE).

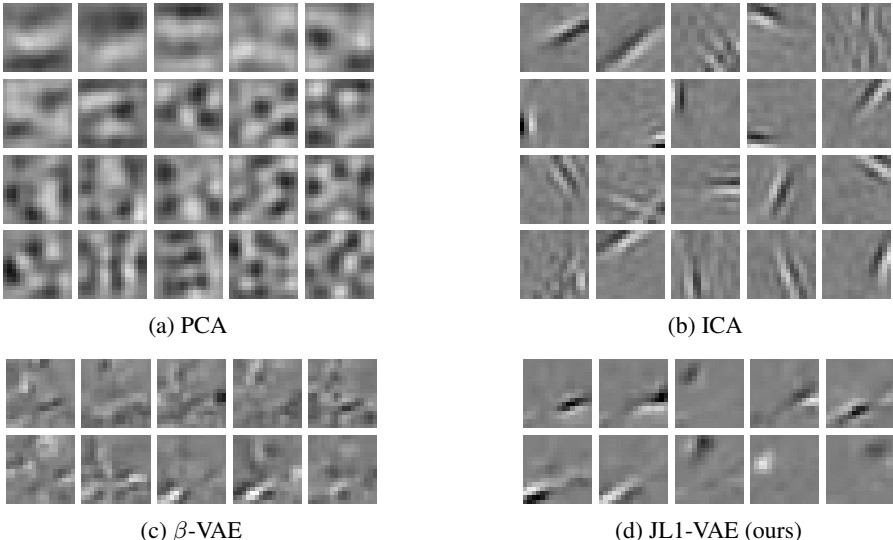

(a) PCA

(b) ICA

(c) $\beta$-VAE

(d) JL1-VAE (ours)

Figure 1: Example columns from generative Jacobian matrices for different modeling techniques on natural image data collected in [7]. ICA used 100 latent dimensions, of which 20 samples are shown. $\beta$-VAE used $\beta = 0.01$; JL1-VAE used $\beta = 0.01$, $\gamma = 0.01$, each on ten latent dimensions.

The use of the generative Jacobian in the JL1-VAE implies a local linear approximation of the generative function, so in order for the JL1-VAE to be useful, the training image data should lie on a manifold [10], and should be sufficiently well sampled. Additionally, JL1-VAE contains inductive

bias, like every other unsupervised disentanglement algorithm [11]. As mentioned above, JL1-VAE's regularization of the generative Jacobian encourages small changes in latent values to result in sparse (impacting a small number of pixels) changes to the resulting image, similar to local receptive fields. This inductive bias is well suited for disentangling motions of different objects in an image, but would presumably not be useful for whole-image changes, such as rotation of the entire image or brightness changes across the whole image.

We apply our novel JL1-VAE framework to a variety of datasets, giving qualitative and quantitative results showing that our added $L_1$ cost can encourage local alignment of the axes of the latent representation with individual factors of variation.

## 2 Background

We present a brief overview of VAEs and introduce the notation we will use throughout this paper. VAEs [1] train a model to generate a data distribution that approximately matches the distribution of unlabeled training data. For the VAEs we consider in this paper, generating a datapoint $\tilde{x} \in \mathbb{R}^n$ from a trained VAE consists of sampling a latent variable $z \in \mathbb{R}^l$ from a standard multivariate Gaussian distribution $N(\mathbb{0}, \mathbb{1})$ and applying a generator function $g : \mathbb{R}^l \to \mathbb{R}^n$ to $z$ to map the latent variable to a generated image $g(z)$. Around this generated image we assume a Bernoulli probability of similar images $\tilde{x} \sim p(\tilde{x}; g(z))$ with the generated image $g(z)$ as its mean.

To train the VAE we also define a multi-variate Gaussian embedding distribution $q(z|x)$ for each training image $x$ with mean $h(x)$ (using a learned embedding function $h : \mathbb{R}^n \to \mathbb{R}^l$) and diagonal covariance $\Sigma_{z|x}(x)$ (using a learned covariance function $\Sigma_{z|x} : \mathbb{R}^n \to \mathbb{R}^l$ that computes the diagonal elements). This embedding distribution $q(z|x)$ is motivated by a desire to approximate the posterior distribution $p(z|x)$.

The objective during training of the VAE is to maximize the Evidence Lower BOund (ELBO), which is defined as $\sum_x L(x)$, where, for each data point $x$,

$$L(x) = E_{z \sim q(z|x)}\left[p(x; g(z))\right] - \text{KL}\left(q(z|x) \| N(\mathbb{0}, \mathbb{1})\right) \tag{1}$$

The ELBO is a lower bound on $\log(p(x)) = \log(\int_z p(x|z)p(z)dz)$, which is the likelihood of the data point given our model. Thus, maximizing the ELBO is a proxy for maximum likelihood estimation. The $\beta$-VAE [2] is an extension to the VAE that multiplies the second term in the ELBO by an adjustable hyperparameter $\beta$. The first term in the ELBO is a stochastic reconstruction loss, so we sometimes refer to $\Sigma_{z|x}$ as the "embedding noise," since it adds noise to the embedding when estimating the stochastic reconstruction loss.

Kumar and Poole [4] show that the stochastic reconstruction term of the ELBO, $E_{z \sim q(z|x)}\left[\log(p(x; g(z)))\right]$, can be approximated using a second-order Taylor expansion as:

$$E_{z \sim q(z|x)}\left[\log(p(x; g(z)))\right] \approx \log p(x|h(x)) + \frac{1}{2}\text{tr}\left(J_g(h(x))^\top H_{p_x}(g(h(x)))J_g(h(x))\Sigma_{z|x}(x)\right) \tag{2}$$

$J_g$ is the Jacobian of the generator function, and $H_{p_x}(g(h(x)))$ is the Hessian with respect to $g(z)$ of the log of the generative probability $\log(p(x; g(z)))$ evaluated at $g(h(x))$. For standard VAE implementations using diagonal Gaussian posteriors and pixel-factorized generative probabilities, $\Sigma_{z|x}(x)$ and $H_{p_x}(g(h(x)))$ are both diagonal.

Equation 2 shows that the stochastic reconstruction loss of the ELBO can be approximated by a deterministic reconstruction loss with a (weighted) $L_2$ regularization cost on the Jacobian $J_g(h(x))$. Kumar and Poole [4] use this weighted $L_2$ regularization in the approximation above to show how the ELBO encourages local alignment of the right singular vectors of $J_g$ to $\Sigma_{z|x}(x)$. That is, larger values of $\Sigma_{z|x}(x)$ (larger embedding noise) in some directions leads to larger $L_2$ regularization on the generator Jacobian in those directions. Equivalently, in directions with large changes in the generator Jacobian, the regularization encourages smaller embedding noise (more precision) in the posterior $\Sigma_{z|x}(x)$. For this work, we use the presence of an implicit $L_2$ loss on the generator Jacobian as further motivation for our choice to add an explicit $L_1$ regularization to the generator Jacobian.

# 3 Related Work

There are several areas of related work we wish to call to the reader's attention. We note previous work in the analysis of the disentanglement properties of $\beta$-VAEs. We delve into previous uses of the term "sparse VAE," as there are (at least) two other common and well studied meanings of "sparse VAE," which we disambiguate from the type of sparsity we study in this work. We note previous VAE work inspired by ICA. We discuss architectural choices that have been shown to improve disentanglement. Finally, we discuss modifications to the VAE objective that have previously been studied to improve disentanglement.

**Disentanglement of $\beta$-VAEs** Mathieu et al. [12] and Rolinek et al. [3] show that restricting the posterior covariance to diagonal (called the mean-field assumption) breaks the rotational symmetry of $\beta$-VAEs. Rolinek et al. [3] further show that this encourages the columns of the generator Jacobian to be orthogonal, relating local $\beta$-VAE latent directions with PCA decomposition. In our work, we take this a step further, explicitly regularizing the sparsity of the generator Jacobian, to break rotational symmetry between directions with equal singular values in of the generator Jacobian.

**Sparse VAEs: Sparsity in VAE codes** Some previous work involving $L_1$ regularization and VAEs uses the term "sparse VAEs" to refer to sparsity in the *latent values* taken on by the latent codes themselves. That is, these works attempt a minimization of something like $\|z\|_1$. This meaning is studied in [13–15]. When we use sparsity in the present work, however, we are not concerned with the values of the latents $z$, but rather with how small axis-aligned changes in the latent values affect the output. That is, we are concerned with sparsity of $J_g(z)$, not $z$. Our associated cost is $\|J_g(z)\|_1$. Our use of "sparsity" has to do with local disentanglement, rather than sparsity in latent values, which is a type of global disentanglement.

**Sparse VAEs: Sparsity in Network Weights** Likewise, other work involving $L_1$ regularization and VAEs (and neural networks more broadly) uses "sparsity" to refer to the desire to make many network weights 0, with the motivation of reducing the size of the stored neural network architecture. This meaning is seen in [16, 17]. In this work, by contrast, we are interested in sparsity in the generator Jacobian, not in the network weights. We note the distinction between sparsity of individual network weights and sparsity in the Jacobian of the overall function. Multiplying sparse matrices does not necessarily result in sparse matrices, and non-linear activations can allow a sparse Jacobian even if the network weights themselves are not particularly sparse. Thus, the $\|J_g(z)\|_1$ cost we study in this work is not what is referred to in studies of sparsity of neural networks, which consider regularizations like the $L_1$ cost over network weights.

**ICA within the VAE Literature** Independent Component Analysis (ICA) [5] is often mentioned in the VAE literature in relation to the role ICA has played in the theory of identifiability and disentanglement of representations [4, 12, 3, 18, 19]. Stuehmer et al. [20] propose using a structured, rotationally asymmetric prior to encourage disentanglement in the embedding. This, and other approaches that attempt to globally match the embedding distribution to a desired shape, are very different from the local, Jacobian-based approach we take in this paper. Khemakhem et al. [21] relate nonlinear ICA with VAEs in the case where the data has an additionally observed variable, and they give a proof that in that case their model is identifiable and correctly disentangles the ground-truth factors of variation. We focus on fully unsupervised training data and assume that we are not given access to any data labels. We are not aware of any prior work applying a sparsity cost to the generator Jacobian, which is the inspiration we take from ICA.

**Architectures shown to improve disentanglement** Previous work has shown impressive results from modifying the network architecture in order to explicitly represent multiple objects by, for example, learning object masks [22], or by modifying how the latent variable is read in to the generative model architecture [23]. Ainsworth et al. [24] and Khan et al. [25] both construct model architectures that explicitly encourage sparsity in the generative network weights. In this work, we focus on how we can regularize the objective function to improve disentanglement, rather than studying how different network architectures can improve disentanglement.

**Modification of VAE prior** Previous work has also investigated modifications to the unit Gaussian prior commonly used in VAEs. Tomczak and Welling [26] use a learnable Gaussian mixture prior.

Stuehmer et al. [20] use a generalized Gaussian distribution (that is not rotationally invariant) as the prior. Bauer and Mnih [27] use rejection sampling to form a more complicated, non-rotationally-invariant prior. Davidson et al. [28] and Perez Rey et al. [29] even modify the prior to lie on non-Euclidean surfaces. Kim and Mnih [18] do not explicitly enforce a prior distribution, but rather use a regularization term to encourage the prior distribution $q(z)$ to be a factorized distribution. While these approaches pressure the entire embedded distribution to have certain properties, we are instead focused on how to modify the learning objective to give *local* bias toward disentanglement, rather than using more global methods based on the overall distribution of the embedded dataset.

**Regularization of VAEs**    Several previous works explicitly or implicitly use $L_2$ normalization of the generator Jacobian [4, 30–32]. Chen et al. [33] regularize by $\|J_g^\top(z)J_g(z) - c\mathbb{1}\|_2$ for some constant $c$. We are not aware of previous investigations of $\|J_g(z)\|_1$ regularization for VAEs.

## 4    Model Loss Calculation and Architecture

### 4.1    Loss Calculation

We define a Jacobian $L_1$ Regularized Variational Auto-Encoder (JL1-VAE) as a VAE that is trained using the $\beta$-VAE loss augmented with an $L_1$ regularization of the Jacobian matrix of the map from latent values to mean generated images. The regularization term is modulated by a hyperparameter $\gamma$.

Specifically, the maximization objective for the JL1-VAE is the sum over all datapoints $x$ of

$$L_{\text{JL1}}(x) = E_{z \sim q(z|x)}\left[ \log p(x|z) - \gamma\big|\big|J_g(z)\big|\big|_1 \right] - \beta\text{KL}\left( q(z|x)\|N(\mathbb{0}, \mathbb{1}) \right) \tag{3}$$

Since we use a Gaussian posterior $q(z|x) = N\left(h(x), \Sigma_{z|x}(x)\right)$, we can use an explicit calculation for the KL-divergence. We estimate the expectation of $\log p(x|z)$ and of $\gamma\|J_g(z)\|_1$ using a single sample $z$ from the distribution over which we are taking the expectation. We estimate the full Jacobian matrix $J_g(z)$ using the finite difference method along each latent dimension. This leads to a runtime that scales roughly linearly with the number of latent variables in the VAE architecture.

### 4.2    Architecture

We use a convolutional architecture for our VAEs. In particular, our embedding architecture consists of convolutional layers followed by a fully connected layer with ReLU activations. This base model is shared between the mean and log variance embedding networks. Each embedding network then appends its own linear fully connected head to the shared model. We use a diagonal structure for the log variance estimates to reduce the number of parameters we need to estimate. We use a latent dimension of ten for all experiments, though we have seen similar results for other latent dimension sizes. The reconstruction architecture consists of fully connected layers followed by convolutional layers, using ReLU activations, with a final sigmoidal activation function. A full set of hyperparameter choices for each experiment can be found in the Appendix. When we compare JL1-VAE with other methods, we ensure consistent architecture choices.

## 5    Experiments

### 5.1    Datasets

To evaluate the ability of JL1-VAE to locally disentangle factors of variation, we apply it to a variety of datasets.

The first are natural images in grayscale taken by Olshausen and Field [7] and cropped to $16 \times 16$-pixel regions. We do not have labeled "ground-truth factors of variation" for this data, but we are able to provide qualitative results by inspecting the columns of the generator Jacobian matrix. This data was made publicly available without a specific license, so we analyze it under fair use.

The second is a dataset of simulated $64 \times 64$-pixel grayscale images of three black dots on a white background, inspired by [34]. The ground-truth factors of variation for this dataset are the x/y

coordinates of the dot centers. We re-implement Zhao et al. [34]'s code to generate the dot images and modify the code so that dots can overlap. We note for this dataset that if a model were to disentangle individual dot motions into different latent directions, then, by symmetry, we would expect identical singular values of $J_g$ for those directions. Thus, we expect that for this dataset $\beta$-VAEs will be unable to isolate individual dot motions, since it has trouble disentangling directions in which the generator Jacobian has equal singular values.

Finally, we also apply our approach to tiled images of a real robotic arm taken from the MPI3D-real dataset [35], licensed under Creative Commons Attribution 4.0 International License. For each data point, we downsample four random images of the robot arm holding a large blue square in different locations and tile the random images in a $2\times2$ pattern to generate a new, more complicated $64\times64$-pixel image containing four different images of a real robotic arm. We call this tiled image dataset MPI3D-Multi.

## 5.2   Training

For the three-dots and MPI3D-Multi datasets, we train using a Bernoulli loss on batches of 64 images over a total of 300,000 batches. We use the Adam optimizer with a learning rate of 0.0001 (matching [11]). We use linear annealing from 0 to the final hyperparameter value over the first 100,000 batches for both the beta hyperparameter and JL1-VAE's $\gamma$ parameter in our implementations for JL1-VAE and $\beta$-VAE (unlike [11]). We note annealing to be beneficial to avoid model collapse when adding our $L_1$ regularization term. We train these models on a Nvidia Quadro V100 hosted locally and one hosted on Google Cloud. Training each JL1-VAE model on ten latent variables takes approximately 2.5 hours, while training each $\beta$-VAE model takes approximately 45 minutes. In total, training ten JL1-VAE models and ten $\beta$-VAE models for quantitative evaluation takes roughly 33 hours.

For the natural image dataset, we train for 100,000 batches of 128 images. We use the Adam optimizer with a learning rate of 0.001 and train on a Nvidia Quadro V100s hosted locally. Training takes nine minutes for the $\beta$-VAE and 23 minutes for the JL1-VAE.

## 5.3   Evaluation Metrics

There are several metrics commonly used to measure "disentanglement" of a latent representation. In this work we address two common metrics, the Mutual Information Gap (MIG) [36] and Modularity [37], and show how we are able to provide extensions to these metrics that give a measure of how well a representation *locally* disentangles factors of variation.

The original MIG and modularity metrics measure global disentanglement—that is, they measure across the whole dataset how well each latent variable maps to a unique ground-truth factor of variation. Since the JL1-VAE does not add an explicitly global disentanglement incentive to $\beta$-VAEs, but instead is designed to locally encourage disentanglement using the Jacobian of the generative map, we do not expect it to necessarily improve the global disentanglement of factors of variation. For example, the JL1-VAE may assign factors of variation to latent variables using one pairing in one local region of the latent space and a different pairing in a different region of the latent space. This could lead to good average local disentanglement, but would not lead to good global disentanglement.

We are therefore interested in defining *local* disentanglement metrics based on MIG and modularity. We call these metrics "local MIG" and "local modularity," and the general form of their calculation is to compute each metric several times on different random "local" samples from the global dataset and then average the results. As with MIG and modularity, the calculation of these disentanglement metrics requires a generative model of the data from ground-truth factor values.

The key technique for each of our local metrics is to repeatedly compute the disentanglement metric on random local samples of data. To generate a random local sample of data, we we randomly choose a centroid from the ground-truth factor values and then random sample N ground-truth datapoints within an $L_\infty$ distance $\rho$ from that centroid. The radius $\rho$ is a hyperparameter determining how close ground-truth factors of variation need to be in order to be considered "local." We scale the hyperparameter $\rho$ as a fraction of the total range of available values for each latent variable. This set of N datapoints comprises each local data sample. For our experiments we choose $N = 10,000$.

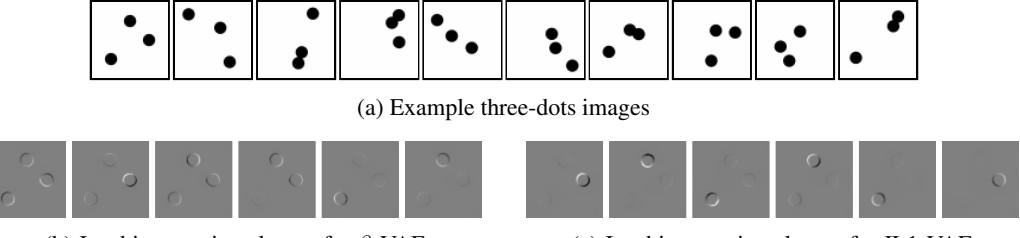

(a) Example three-dots images

(b) Jacobian matrix columns for $\beta$-VAE          (c) Jacobian matrix columns for JL1-VAE

Figure 2: Qualitative results for the three-dots dataset. We show six Jacobian matrix columns for $\beta$-VAE ($\beta = 4$) and JL1-VAE ($\beta = 4$, $\gamma = 0.1$) evaluated for the leftmost example image.

For each local sample of data, we apply the MIG and modularity metrics to that sample to determine the disentanglement of the latent space in that local region. We use the MIG and modularity calculation implementations from the open-source (Apache License 2.0) `disentanglement_lib` library [11].

We repeat this algorithm with 20 different local data samples and report the average as the local disentanglement score.

## 6   Results

We present qualitative results for the three-dots, MPI3D-Multi, and natural image datasets, and quantitative results for the three-dots dataset.

### 6.1   Qualitative Results

Qualitative results are generated by inspecting the generator Jacobian at the (deterministic) latent embeddings for example images. Each generative Jacobian matrix column is associated with a latent direction and shows how the generated image would change from a slight perturbation to the embedding in that latent direction. The generative Jacobian matrix columns for natural images are shown in Figure 1. There, we show the largest 20 PCA components, an arbitrary sample of 20 ICA components (from training 100 ICA components using FastICA [38]), and the ten Jacobian matrix columns for the $\beta$-VAE and JL1-VAE models. Additional visualizations are included in the Appendix. We discern more localization (results more similar to local receptive fields) in the JL1-VAE and ICA results, compared to the $\beta$-VAE and PCA results.

For the three-dots dataset and MPI3D-Multi, we follow the same procedure to generate qualtitative results. We show results for the three-dots dataset in Figure 2. There, we show the six Jacobian columns with the largest $L_2$ norms for the three-dots dataset for both our JL1-VAE and $\beta$-VAE (all ten Jacobian columns are visualized in the Appendix). Qualitatively, we see that, when evaluating the Jacobian of the generator function for our JL1-VAE, individual dot motions are separated into different latent components. The $\beta$-VAE does not exhibit this behavior.

For the MPI3D-Multi dataset, containing tiled images of a real robot, we again see that the JL1-VAE does a better job separating the four robots contained in each image into separate latent variables. These results are presented in Figure 3.

### 6.2   Quantitative Results

We generate local disentanglement scores for the models trained on the three-dots images.

To observe the effect of the $\rho$ parameter of our local disentanglement metrics, in Figure 4. we plot the varying local disentanglement scores as we change the $\rho$ parameter for our JL1-VAE ($\beta = 4.0$ and $\gamma = 0.1$) and a standard $\beta$-VAE ($\beta = 4.0$). The hyperparameter for $\beta$ was chosen near the middle of the range used in [11]. We also tried other hyperparameter values and saw similar results. Too large a $\gamma$ can lead to model collapse, so we chose a small enough $\gamma$ to avoid that collapse but otherwise large enough to start to see reconstruction performance degradation, so we knew that its regularization was affecting model training.

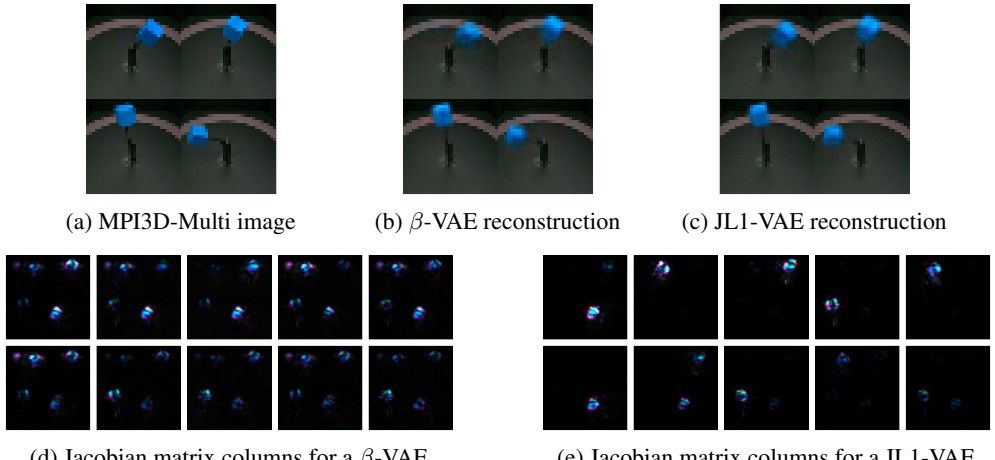

(a) MPI3D-Multi image      (b) $\beta$-VAE reconstruction      (c) JL1-VAE reconstruction

(d) Jacobian matrix columns for a $\beta$-VAE      (e) Jacobian matrix columns for a JL1-VAE

Figure 3: Qualitative results for MPI3D-Multi. JL1-VAE shows stronger pressure to locally disentangle individual robot motions. Both models used $\beta = 0.01$. For JL1-VAE, $\gamma = 0.01$.

We note quantitatively that JL1-VAE attains higher local disentanglement scores compared to $\beta$-VAEs, which is especially true as we look at more localized samples of data, corresponding to a smaller $\rho$ parameter. For $\rho = 0.1$ we see significantly higher local disentanglement scores for JL1-VAE compared to $\beta$-VAE ($p < 0.001$ for T-test), but for $\rho = 1$, testing for global disentanglement, we see indistinguishable disentanglement scores between the two ($p > 0.05$ for T-Test). Our JL1-VAE is able to *locally* disentangle factors of variation for the three-dots dataset, but does not globally disentangle factors of variation.

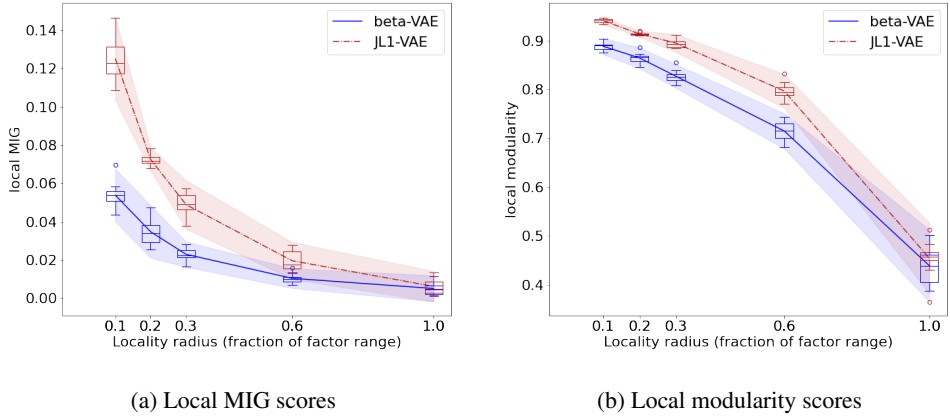

(a) Local MIG scores      (b) Local modularity scores

Figure 4: Local disentanglement scores varying the locality parameter $\rho$. Ten JL1-VAE and $\beta$-VAE models were trained on the three-dots dataset with $\beta = 4$ and, for JL1-VAE, $\gamma = 0.1$.

Fixing the $\rho$ parameter to $0.1$, we also compute the local MIG and local modularity scores for six different comparative methods, namely $\beta$-VAE, FactorVAE, DIP-VAE-I, DIP-VAE-II, $\beta$-TCVAE, and AnnealedVAE, using the implementations of `disentanglement_lib` with our convolutional architecture. A description of each of these models can be found in [11]. We trained ten iterations of those models with different random seeds using hyperparameters chosen near the middle of the suggested ranges in that work. That included training ten new $\beta$-VAE models with new random seeds. All models were trained with ten latent dimensions.

Local MIG and local modularity scores are shown in Figure 5. We see a range of disentanglement scores due to the random seeds used to generate our models (ten models for each learning algorithm). Additionally, our local MIG and modularity metrics have some additional stochasticity due to the randomness in sampling 20 local samples of 10,000 points during calculation of those metrics.

Nevertheless, we observe significantly higher disentanglement scores ($p < 0.001$ for T-test) for our JL1-VAE compared to every baseline method.

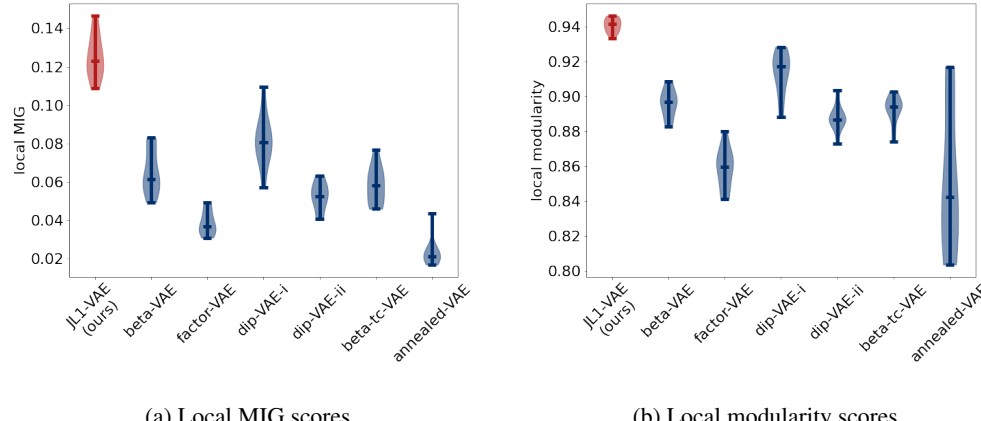

(a) Local MIG scores           (b) Local modularity scores

Figure 5: Local disentanglement scores for JL1-VAE models and baseline implementations from [11]. The baseline implementations use default hyperparameters from that paper, choosing values near the middle when a range of hyperparameters are listed. Each model is run ten times with new random seeds. Local disentanglement is calculated using $\rho = 0.1$ with 20 different local samples.

## 7   Discussion

In this work, we presented JL1-VAE, a VAE augmented with an $L_1$ regularization to the Jacobian to improve local disentanglement. We extended the MIG and modularity disentanglement metrics to generate metrics that can measure local disentanglement. We evaluated our model on natural images, simulated images of dots, and tiled images of a real robot, and showed qualitatively and quantitatively that our method can improve local disentanglement in the generated representation.

Our added $L_1$ regularization to the Jacobian of the generator function is motivated by the use of $L_1$ regularization to prefer certain orientations in ICA and sparse coding, by a desire to relate each latent direction to sparser pixels (more similar to localized receptive fields), and by the implicit $L_2$ regularization already present in $\beta$-VAEs. We show that this $L_1$ regularization term can encourage latent axes to locally align with ground-truth factors of variation. While this approach shows promise for local alignment, it does not address global alignment issues. That is, in one part of the dataset, the learned representation may assign a latent variable $e_1$ to follow a certain latent factor of variation, and in a different part of the dataset it might be a different latent variable $e_2$ that follows that latent factor of variation.

Regarding "no free lunch" theorems that show unsupervised disentanglement is impossible without inductive biases [11], we note that $L_1$ regularization of the generative Jacobian generates an inductive bias. The inductive bias encourages small axis-aligned perturbations of the latent space to result in sparse changes to the image space, whether that be expressed as localized receptive fields which act only on small regions of the image space, as seen in Figure 1, or motions of only a single dot or robot, as seen in Figures 2 and 3. Cases where this inductive bias might not add value would include, for example, single robot images from MPI3D, where the primary ground-truth factors of local variation affect the same object within the same image patch, or images where one factor of variation includes global lighting changes.

This work has only applied this learning method to image data, and we would like to apply this approach to multimodal data gathered from a real robotic system to understand broader applications.

Finally, this approach computes the full Jacobian of the generator during training, leading to training times that scale linearly with the number of latent dimensions. Future work should look to sampling-based methods to approximate the $L_1$ cost by only computing a subset of the Jacobian values in order to speed up training.

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
