# A  Neural Network Architecture

We use a convolutional neural network architecture for our models. Our code can be found in our open source repository[1]

For the auto-encoders used on 64×64-pixel images, we mimic the architecture presented in [11]. We use 4×4 kernels for all convolutional layers with a stride of 2. We use a ReLU between all layers, with a final sigmoidal layer on the reconstruction architecture and a Bernoulli loss. In Tables 1 and 2, "Conv2d" refers to a convolutional layer, "FC" refers to a fully connected layer, "ConvT2d" refers to convolutional transpose, and the "(× 2)" in the embedding architecture refers to the separate mean and log variance heads on the shared architecture.

Table 1: Embedding and reconstruction architectures for 64×64-pixel images

| Embedding | Reconstruction |
|---|---|
| Input: 64×64, 1 or 3 channels | Input: 10 values |
| Conv2d: 32 channels | FC: 256 channels |
| Conv2d: 32 channels | FC: 4×4 image, 64 channels |
| Conv2d: 64 channels | ConvT2d: 64 channels |
| Conv2d: 64 channels | ConvT2d: 32 channels |
| FC: 256 channels | ConvT2d: 32 channels |
| FC (× 2): 10 values | ConvT2d: 64×64, 1 or 3 channels |

For the auto-encoders used on 16×16-pixel images (the natural image crops), we use 3×3 kernels for all convolutional layers and a stride of 2 everywhere except for the last reconstruction layer, which has a stride of 1. We use a ReLU between all layers, with a final sigmoidal layer on the reconstruction architecture and a Bernoulli loss.

Table 2: Embedding and reconstruction architectures for 16×16-pixel images

| Embedding | Reconstruction |
|---|---|
| Input: 16×16, 1 channel | Input: 10 values |
| Conv2d: 64 channels | FC: 128 channels |
| Conv2d: 128 channels | FC: 4×4 image, 64 channels |
| FC: 128 channels | ConvT2d: 64 channels |
| FC (× 2): 10 values | ConvT2d: 32 channels |
| | ConvT2d: stride 1, 64×64, 1 channel |

For the loss, We estimate the full Jacobian matrix $J_g(z)$ using the finite difference method along each latent dimension. That is, for any given latent value $z$ at which we wish to compute the Jacobian matrix, we generate a set of $k$ data points $z_i = z + \epsilon e_i$, where $\epsilon$ is a small fixed value and $e_i$ a unit vector in the $i^{\text{th}}$ latent direction. We then run the forward model on the batch of $z_i$ to generate $g(z_i)$ and estimate the $i^{\text{th}}$ column of the Jacobian matrix as $(g(z_i) - g(z))/\epsilon$ This Jacobian matrix estimate is itself backward differentiable using standard backward differentiation, and can be directly used in our JL1-VAE loss.

# B  Three-dots Experiment Hyperparameters and Additional Results

For the three-dots dataset, We discretize the possible x,y coordinates of the center of each dot to 64 different values. We note that the generative map is not injective, as the dots are identical, so the same resulting image can be formed from multiple permutations of ground-truth factor values. There are $64^6 \sim 68.7$ billion different possible input latent combinations, from which we pre-generate a cache of 500,000 images on which we train. During evaluation, we generate new images at runtime based on the desired ground-truth factors of variation.

---

[1]https://github.com/travers-rhodes/jlonevae




(a) Jacobian matrix columns for $\beta$-VAE          (b) Jacobian matrix columns for JL1-VAE

Figure 6: Qualitative results for three-dots. Both models used $\beta = 4.0$. For JL1-VAE, $\gamma = 0.1$. All Jacobian matrix columns are shown.

We train a $\beta$-VAE with $\beta = 4$ on a training dataset cache of 500,000 64×64-pixel images of three black dots on a white background, with $x$ and $y$ values for the dot centers appearing independently at one of 64 possible discrete locations, evenly spaced horizontally and vertically, across the image. We embed the dataset into a latent space of 10 dimensions. We train on 300,000 independently sampled batches of 64 images from the cache, giving a total of 19,200,000 image presentations to the neural network. Additionally, we train our JL1-VAE with the same $\beta$ and model architecture on the same training dataset with our added $L_1$ regularization weighted by a hyperparameter $\gamma = 0.1$. The hyperparameter $\gamma$ was chosen as the largest tested for which the learning algorithm converged to give good reconstruction accuracy. We use linear annealing for both the $\beta$ and $\gamma$ parameters, annealing each from 0 to their final values over the first 100,000 batches. We use the Adam optimizer with a learning rate of 0.0001.

For the baseline comparison models, we use the implementations of $\beta$-VAE, FactorVAE, DIP-VAE-I, DIP-VAE-II, $\beta$-TCVAE, and AnnealedVAE from [11], matching their hyperparameter choices. For implementations that for which they provided a range of tested hyperparameters, we chose near the middle of their range. Thus, for $\beta$-VAE we used $\beta = 4$; for Annealed VAE we use $c_{max} = 25$, iteration threshold$= 100000$, and $\gamma = 1000$; for Factor VAE we use $\gamma = 30$; for DIP-VAE-I we use $\lambda_{od} = 5$ and $\lambda_d = 50$; for DIP-VAE-II we use $\lambda_{od} = 5$ and $\lambda_d = 5$; and for $\beta$-TCVAE we use $\beta = 4$.

We note that we modified the reference implementation provided with that work from in order to have consistent 4×4 kernels as shown in the architecture in Table 1. We include the modified implementation in our supplemental materials. The reference implementation of that architecture had unexplained 2×2 convolutional kernels for two layers.

We varied the random model initialization seed ten times and trained ten different models for each algorithm type. Additionally, we ran a smaller experiment randomizing both the model initialization seed and using different initial seeds for data sampling as well, getting similar results to those presented in the paper. The baseline implementation samples batches by epoch, shuffling after each epoch, while our implementation pulls independent random batches at each training set. Thus, the results for $\beta$-VAE in Figure 4 use independently-sampled random batches, while the results for $\beta$-VAE in 5 use shuffling after each epoch. This did not seem to affect results.

For the local metric calculations, we use the implementation provided by [11]. We sample 20 different local regions, pulling 10,000 points for each. We use a histogram discretization with 5 bins for mutual information calculations.

All ten Jacobian columns associated with Figure 2 are shown in Figure 6.

Additionally, we validate that an $L_2$ loss does not have the same disentangling properties as our $L_1$ loss by replacing the $L_1$ loss with an $L_2$ loss and computing the local disentanglement metrics in Figure 7. We label the JL1-VAE with $L_1$ replaced by $L_2$ a JL2-VAE. All the $L_2$ regularizations are roughly indistinguishable from the $\beta$-VAE result, while the JL1-VAE consistently outperforms for the full range of tested regularization values $\gamma$.

## C   Natural Image Experiment Hyperparameters and Additional Results

We train a $\beta$-VAE with $\beta = 0.01$ on a dataset 100,000 16x16-pixel crops from grayscale natural scenes [7], embedding the dataset into a latent space of 10 variables. We train for 100,000 batches of 128 images, re-shuffling the images after each epoch. We note that due to epoch endings a few of the batches were incomplete, with fewer than 128 images. Additionally, we train our JL1-VAE with the

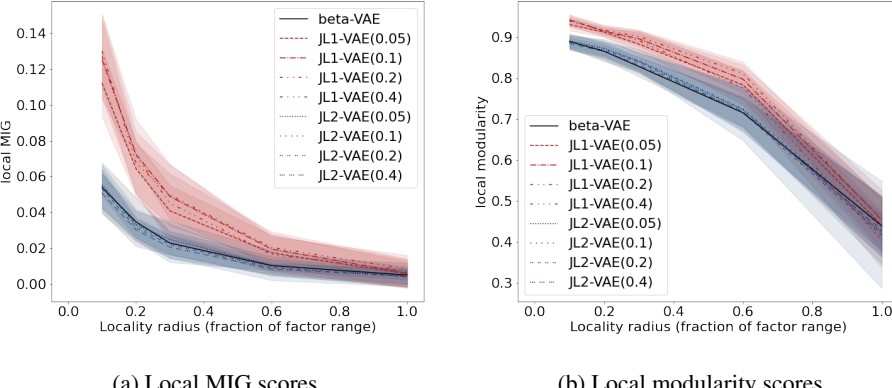

(a) Local MIG scores          (b) Local modularity scores

Figure 7: Local disentanglement scores varying the locality parameter $\rho$ and the regularization factor $\gamma$ for JL1-VAE, JL2-VAE, and $\beta$-VAE. The regularization factor $\gamma$ is given in parentheses in the legend. Ten of each type of model were trained on the three-dots dataset with $\beta = 4$. This figure best viewed in color.

same model architecture and $\beta$ on the same training dataset with our added $L_1$ regularization cost weighted by a hyperparameter $\gamma = 0.01$. The $\beta$ was chosen as large as possible that still avoided significant dimensionality collapse, and then the hyperparameter $\gamma$ was chosen as the largest tested for which the learning algorithm converged to give good reconstruction accuracy. We use linear annealing for both the $\beta$ and $\gamma$ parameters, annealing each from 0 to their final values over the first 50,000 batches. We use the Adam optimizer with a learning rate of 0.001.

We show additional Jacobian column results, training on five latent dimensions in Figure 8, and training on 25 latent dimensions in Figure 9. We also show the top 100 PCA components and 100 trained ICA components in Figure 10.

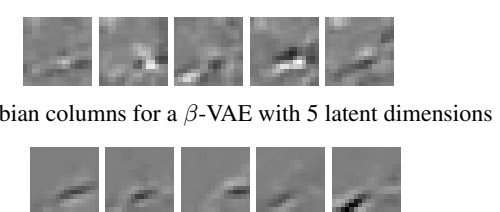

(a) Jacobian columns for a $\beta$-VAE with 5 latent dimensions

(b) Jacobian columns for a JL1-VAE with 5 latent dimensions

Figure 8: Results for $\beta$-VAE and JL1-VAE using 5 latent dimensions (instead of the 10 shown in the main paper). Both are trained with $\beta = 0.01$, and JL1-VAE trained with $\gamma = 0.01$.

## D  MPI3D-Multi Experiment Hyperparameters

We train a $\beta$-VAE with $\beta = 0.01$ on a 2×2 tiling of every-other-pixel downsampling of randomly sampled images pulled from MPI3D-real, resulting in 64×64-pixel training images. We only sample MPI3D-real images of a top-down view of the robot holding a large, blue cube, with salmon background lighting. In this way, our dataset does not vary along unordered/sparse latent factors like color/shape. This leaves two independent dimensions of variance (horizontal and vertical axis joints) for each of the 4 tiled robot images. If we were to apply our local disentanglement metrics to discrete factors of variation that do not come with a natural distance metric such as "shape" or "color," we would require equality in order for values to be considered "close." That is, for any such factor of variation, a "local" sampled dataset will be constant on that factor.

We embed the dataset into a latent space of 10 dimensions. We train on 300,000 independently sampled batches of 64 images from the cache, giving a total of 19,200,000 image presentations to the neural network. The $\beta$ was chosen to give good reconsruction accuracy. Additionally, we train our

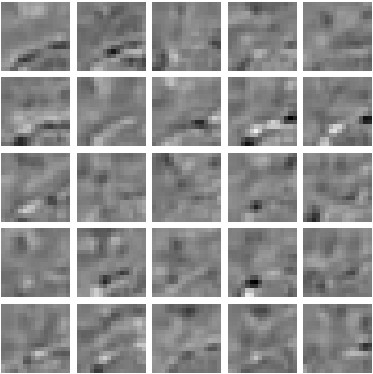

(a) Jacobian columns for a $\beta$-VAE with 25 latent dimensions

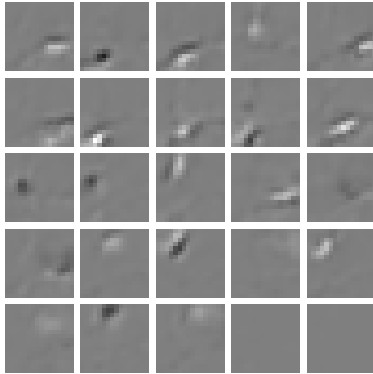

(b) Jacobian columns for a JL1-VAE with 25 latent dimensions

Figure 9: Results for $\beta$-VAE and JL1-VAE using 25 latent dimensions (instead of the 10 shown in the main paper). Both are trained with $\beta = 0.01$, and JL1-VAE trained with $\gamma = 0.01$.

JL1-VAE with the same $\beta$ and model architecture on the same training dataset with our added $L_1$ regularization weighted by a hyperparameter $\gamma = 0.01$. The hyperparameter $\gamma$ was chosen as the largest tested for which the learning algorithm converged to give good reconstruction accuracy. We use linear annealing for both the $\beta$ and $\gamma$ parameters, annealing each from 0 to their final values over the first 100,000 batches. We use the Adam optimizer with a learning rate of 0.0001.

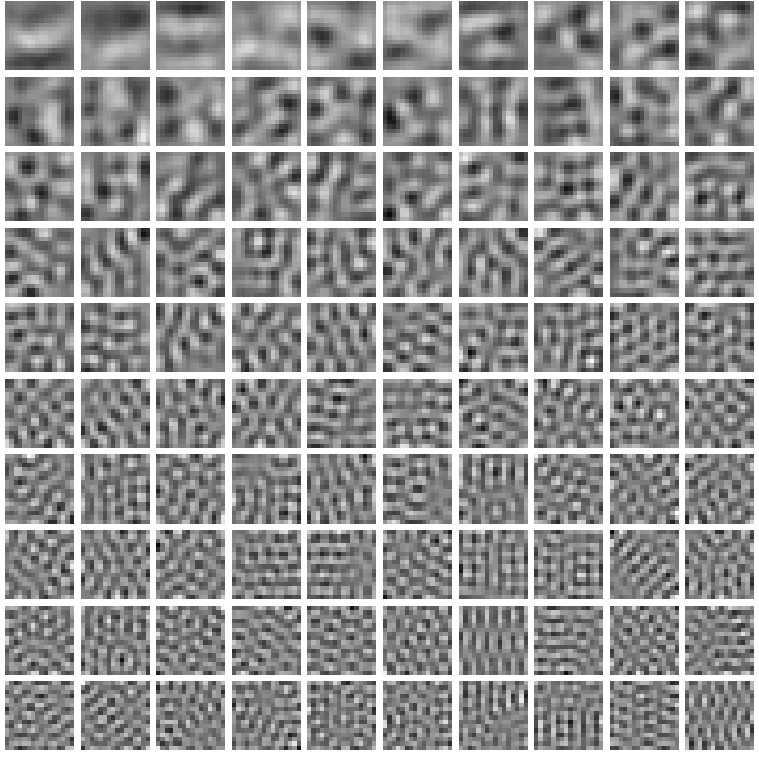

(a) 100 latent PCA directions with the largest explained variance [38]

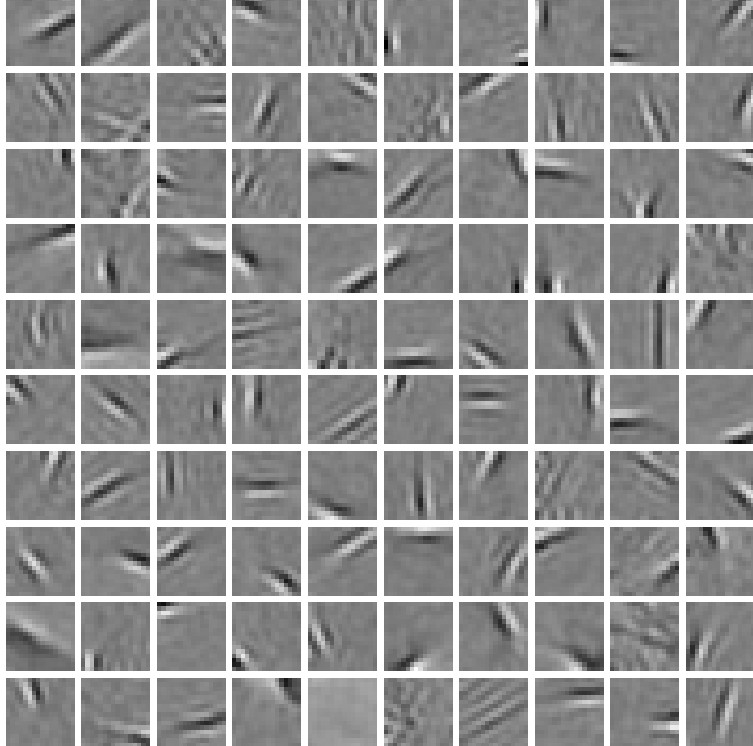

(b) 100 latent ICA directions fit using FastICA[38, 5]

Figure 10: Latent vectors for PCA and ICA trained on random 16x16 crops from natural images collected by [7]