# OpenReview forum: "Local Disentanglement in Variational Auto-Encoders Using Jacobian $L_1$ Regularization"
_NeurIPS.cc/2021/Conference — NeurIPS 2021 Poster_

### Official Review · Reviewer_ZTNa · 2021-07-01

**Rating:** 7
**Confidence:** 4

**Summary:**

The paper presents a novel disentanglement strategy for VAEs based on the induced sparsification of the decoder Jacobian.

**Limitations And Societal Impact:**

Limitations are well defined, aside of that of point 1) above, which I feel is a critical flow, if not addressed properly.

**Main Review:**

The paper addresses an important problem in representation learning and the solution it proposes is simple and, to the best of my knowledge, novel. However, there is a number of issues I found reading the manuscript.

1) Only this one is potentially a critical flaw of the technique and therefore I will start from this. I have some doubt that a sparse decoder Jacobian implicates local disentanglement in the general case. Using the image domain as an example, as done in the paper, a sparse Jacobian means that changes in one element of a latent variable cause changes in the fewest pixels in the generated image. This corresponds to disentanglement only if the generating factors of variations are aligned with this notions, e.g. small objects moving, but it does not for more global factor of variations, e.g. global coloration, shadowing ecc. The experiments in the paper all use datasets that seem to be specifically suited to have this jacobian sparsity-disentanglement correspondence, which I don't think is true in the general case. There would need to be further experiments with more globally affecting sources of variation in the data, or readjust the claims of disentanglement.

2) There may be two typos in the technical section that makes things rather confusing. On line 154: "we show how penalizing the sparsity of the Jacobian, that is, penalizing the L1 norm...". Should it be "encouraging the sparsity"? Otherwise "penalising the sparsity" and "penalising the L1 norm" are contradicting.
Then, in equation 3, should it be a minus sign in front of the gamma*L1_norm term? (or gamma is negative?). This is the ELBO, therefore it is maximised, and I assume you want to minimise the L1 norm.
These might just be unfortunate typos, but in combination make the core definition of your method very confusing.

The reason for my low score is mostly point 1) and I will be happy to higher it if it is convincingly addressed, or alternatively if the claims are readjusted to better defined in which domains we can expect the proposed method to give good disentanglement.

**Time Spent Reviewing:**

3

---

> ### Author Response · Authors · 2021-08-10
> **Author Response**
>
> Thank you for taking the time to read our paper and for your review. In "Challenging Common Assumptions in the Unsupervised Learning of Disentangled Representations" (Locatello et al. 2019), Locatello et al. showed that "unsupervised learning of disentangled representations is fundamentally impossible without inductive biases." Thus, like all unsupervised learning techniques for disentanglement, our method does rely on the inductive biases in the model being useful/relevant to the particular data set.
>
> Our added inductive bias toward spatially localized features is meant to generate the local receptive field-like features seen in ICA/Sparse Coding and in simple cells in mammalian primary visual cortex (Olshausen and Field 1996), so we think that it is good and biologically-inspired to encourage this type of part-based/region-based disentanglement of separate objects in the image. For reasonable regularization values, we expect that other inductive biases already present in $\beta$-VAEs will still be able to mimic other disentanglement properties of the mamallian visual system, like those seen, for example, in "Unsupervised deep learning identifies semantic disentanglement in single inferotemporal neurons" (Higgins et al., 2020).
>
> We intended the discussion paragraph starting on line 337 to explain what you point out in your review---that our model has an inductive bias toward spatially sparse Jacobians and therefore spatially localized features, rather than looking for global changes across the whole image. We already note there that we do not expect this regularization and associated inductive bias to be useful on all datasets, but we will amend our paper to explain the inductive bias better, including your example about global brightness, and, importantly, will also make sure to include that discussion in the introduction, not just in the summary, so that readers are sure to understand that we don't expect this technique to improve disentanglement on all datasets --- only those where our particular inductive bias is useful.
>
> Thank you for bringing to our attention our very unfortunate typos in the core definition of our method. Our implemented objective does indeed _encourage_ sparsity, and our actual implementation does try to minimize the L1 norm term. When writing the optimization objective as a maximization problem we should have written a minus sign before the $\gamma$ times L1 norm term. Thank you again for finding those typos!

---

> > ### Comment · Reviewer_ZTNa · 2021-08-11
> > **Inductive Bias**
> >
> > Thank you for the concise background on the matter. The link to biological observations is a strong motivation to focus on "pixel sparsity" and the discussion over inductive bias would personally be extremely interesting, even though partially already treated by the literature you cited. I trust you will include some key sentences to clarify/explain the points you just explained here, and therefore I will raise the score.

---

### Official Review · Reviewer_Bxq2 · 2021-07-05

**Rating:** 6
**Confidence:** 4

**Summary:**

In this paper, the authors try to solve the identifiability issue of representation learning for generative processes with similar eigenvalues. Inspired by ICA, sparse coding, the authors propose to add an L1 regularizer of the Jacobian of decoder mean-mapping to the ELBO. To compare with other methods, the authors also adapt the common evaluation metrics (including MIG and Modularity) to the problem of local disentanglement. Both qualitative and quantitative results show the efficiency of the proposed method compared with baseline models.

**Limitations And Societal Impact:**

This paper shares the limitations and potential negative social impact of most works on representation learning. As a result, I don't think the authors need to solve these issues in this paper.

**Main Review:**

Pros:
1. This paper discusses the novel problem of local disentanglement, which is interesting and might be applicable to problems such as data manipulation. Also, the solution is rather simple to be used in all VAE-based (even GAN-based) models.
2. The authors provide a very comprehensive survey of related works, including works on disentanglement and sparse VAE.
3. Results shown in Sec.6 are interesting. Particularly, the comparison with beta-VAE in Fig.2 is convincing that the proposed method works.

Cons:
1. The correctness of the proposed method is based on the hypothesis that the mapping from latent features to real data in real generation process is locally linear. Otherwise, the L1 loss learn is useless to recover the real generation process. There is no discussion about this hypothesis and experiments are performed on simple datasets where this hypothesis naturally holds.
2. The ablation study of \rho in Fig.4 make me feel that the authors are selecting the best parameter of the evaluation metric specifically for their proposed method. It might be unfair. Also, the authors might want to add some experiments on downstream tasks to show the proposed method is useful.
3. It is not necessary to introduce how to calculate Jacobian in Sec. 4.1. There are packages to do that. Please refer to https://stackoverflow.com/questions/50244270/computing-jacobian-matrix-in-tensorflow/53798621.
4. Some mathematical mistakes:
  a. In Eqn.2. H should be the Hessian of logP rather than P. Otherwise, the matric is no longer diagonal.
  b. In line 169, I think the i-th column of Jacobian matrix should be (g(z_i)-g(z))/epsilon.

**Time Spent Reviewing:**

4

---

> ### Author Response · Authors · 2021-08-10
> **Author Response**
>
> Thank you for your helpful review. Concerning our assumption that the local linear approximation was reasonable, we will add a discussion on our assumption that we are modeling continuous factors of variation and that we have granular enough data sampling so that a locally linear approximation is reasonable.
>
> With regard to our choice of $\rho$ for Figure 5,  the choice of a small $\rho$ is so that we can measure local, rather than global, disentanglement. This paper is focused on local disentanglement. If we had chosen a very large rho, like $\rho = 1$, then we would be measuring global disentanglement, which is something (as Figure 4 shows) we are not able to achieve with this methodology.
>
> We will streamline our discussion of the Jacobian to simply point out that we are using finite differences to compute our Jacobian instead of an analytic solution (an analytic solution is implemented in the package linked in your comment). For PyTorch, we found that finite difference methods were significantly faster than using an analytic Jacobian calculation.
>
> Also, thank you for catching our mathematical mistakes!

---

> > ### Comment · Reviewer_Bxq2 · 2021-08-27
> > **Thanks for your reply**
> >
> > I thank the authors for their explanation. However, because the authors only responded to some of my questions and provide little new information, the following concern remains,
> > - Fairness of evaluation metric.
> > - Performance on downstream tasks.
> > - Still no discussion about locally linear approximation, only a promise.
> >
> > Though I still think this paper provides a potentially useful method, there is some space for improvement in experiments and discussions.

---

> > > ### Author Response · Authors · 2021-08-31
> > > **Author Response**
> > >
> > > Thanks again for your review.
> > >
> > > Our evaluation metric is the standard MIG metric ("Isolating Sources of Disentanglement in VAEs", Chen et al. 2018) repeatedly applied to random epsilon-ball subsamples of the data with size parameterized by $\rho$. With $\rho=1$ our local MIG is equivalent to the standard (global) MIG disentanglement metric. For $\rho$ small, our metric becomes a measure of local disentanglement.
> > >
> > > We hope to address downstream tasks in future work.
> > >
> > > Smoothness assumptions are common in the image domain ("The Manifold Ways of Perception", Seung and Lee, 2000), and we appreciate the reviewer's reminder to explicitly state this assumption in our exposition.

---

### Official Review · Reviewer_rKED · 2021-07-12

**Rating:** 7
**Confidence:** 4

**Summary:**

Authors propose to regularize the Jacobian of the generator in a VAE setting so as to encourage local alignment of the latent axes, i.e. `local disantanglement'. They do so by finite differences, and empirically show on the MPI3D dataset the effectiveness of their approach.

**Limitations And Societal Impact:**

Yes limitations are discussed.

**Main Review:**


## Strengths
The introduced method is quite well motivated and simple to implement. It empirically appears to be better suited than L2 regularization to disentangle the generative factors. These empirical results a fairly convincing.

## Weaknesses
As stressed in the paper, this method brings improvement in terms of `local' disantanglement but does not address global alignment. This is fair but there is perhaps a lack of motivation on why local disantanglement matters. This is not completely unsurprising as the Jacobian is by definition a local concept.
Regarding computational aspects of the method, the regularization term $\|J_g\|_1$ is computed via with finite difference. This scales linearly in the number of latent dimensions. In practice, is this a marginal cost?
It would be interesting to see how a L2 regularization would perform (as opposed to L1 as proposed), as it is argued to lead to an objective that is invariant to rotation in  subspaces of equal singular vectors. In practice, is it very likely that some singular values be equal?

## Clarity
The paper is fairly well-written. I believe that it would be worth improving the explanation of Equation 2 as it is key in the motivation of the introduced method. Additionally, it is claimed that the approach is inspired by ICA, yet it is not detailed how so, nor is ICA really introduced. Regarding Sections 5 and 6, there are lots of details, yet paradoxically it takes a bit of time to understand the empirical settings. I believe that these sections can be streamlined a bit.

## Relation to prior work
Although the Related Work section is pretty good, I believe that a brief discussion of contractive auto-encoders (Rifai et al., 2011) could be useful as these regularize the $l2$ norm of the Jacobian and show that 'this penalty helps to carve a representation that better captures the local directions of variation dictated by the data'. If space is needed, the paragraph 'Sparse VAEs: Sparsity in Network Weights' could be removed since it is not extremely relevant to the topic (as the paragraph itself it stating).

## Reproducibility
The code source has been provided in the supplementary material and after a quick inspection, it seems to allow to reproduce their results.

## Additional feedback
140: "beta-VAE"
Equation 2: Would be useful to give a bit more context on how this equation is derived and what does the approximation symbol means here, as this is a key motivation for the later L1 regularization. Is this specific to $\beta$-VAE also?
Section 6.2: Wouldn't it be useful to visualize latent interpolations?
Figure 4: May be easier to plot error bars / confidence interval instead of each sample.



**Time Spent Reviewing:**

4

---

> ### Author Response · Authors · 2021-08-10
> **Author Response**
>
> Thank you for your thoughtful review. To answer your questions:
>
> We agree that the time cost for finite difference estimation of the Jacobian, which scales linearly with the number of latent variables, can be problematic when there is a large number of latent dimensions (eg: >100 latent variables), so we believe that future work to reduce this time cost, as suggested in the last paragraph of the paper, would be valuable.
>
> With regard to adding an L2 regularization instead of an L1 regularization, we believe that, as shown in "On Implicit Regularization in $\beta$-VAEs" (Kumar and Poole 2020), the $\beta$-VAE already actually contains an implicit L2 regularization term, so we would argue that our comparison to $\beta$-VAE is already an (implicit) comparison to an L2 regularization.
>
> Thank you for the pointer to relevant related work. We had cited "The Manifold Tangent Classifier" (Rifai et al. 2011) in the related work section "Regularization of VAEs", but we agree that it would make more sense to directly cite the paper that introduced CAE's, rather than the paper we cited (which is a subsequent paper that utilized CAE's).
>
> Also, thank you for pointing out opportunities for us to improve the clarity of our explanations in Equation 2 and Sections 5 and 6. We will improve the explanations in those areas. We'll also switch Figure 4 to a line plot with confidence intervals. For the visualizations in 6.2, we visualize the Jacobian to show that small perturbations to latent values affect different regions (a test of local disentanglement), whereas a standard latent interpolation would sweep the full range of latent values (a test of global disentanglement).
>
> We also appreciate the feedback that we did not make it clear how we were inspired by ICA and Sparse Coding. The main inspiration from ICA/Sparse coding is from the presence of spatially localized receptive fields when ICA/Sparse Coding is trained on natural images. To mimic this, we wanted to encourage sparsity within the Jacobian columns of our model, so that generative factors of variation only affect image patches (few number of pixels), thereby replicating the localized receptive fields seen when ICA is trained on natural images. The other, more minor, inspiration is from the presence of the L1 regularization in (some implementations of) ICA and Sparse Coding, which is a non-rotationally symmetric and sparsity-inducing cost. We will edit the relevant paragraphs starting on line 36 to clarify our inspiration.

---

> > ### Comment · Reviewer_rKED · 2021-08-19
> > **Reply to rebuttal**
> >
> > I thank the authors for the clarifying comments.
> >
> > It is true $\beta-VAE$ actually implicit contains an L2 regularisation. It would be interesting to compare the L1 vs L2 with varying value of the regularisation coefficient. Currently, in Figure 4 or 5 the coefficient is fixed. I believe it may help to understand how and why the lack of rotational invariance of the L1 norm is key.
> >
> > The paper would be strengthen by clarifying the relationship with ICA/Sparse coding and the motivation for spatially localized features.
> > I am adjusting my rating, and hope authors will update accordingly their paper so as to address the remaining issues.

---

### Decision · Program_Chairs · 2021-09-27

**Decision:**

Accept (Poster)

**Comment:**

This paper introduces a simple and effective way to disentangle factors of variation in VAE representations by penalizing the Jacobian of the decoder with an L1 regularizer. The reviewers unanimously agree that this is a compelling approach, and their concerns were addressed during the discussion period. There are a number of clarifications that should be added to the final draft based on these discussions, but perhaps the main one is to clarify the relationship between the approach and ICA/sparse coding based on learning localized features. Please read through the reviews carefully and ensure that the paper is updated accordingly, as requested by the reviewers.